# Performance of Two Extracts Derived from Propolis on Mature Biofilm Produced by *Candida albicans*

**DOI:** 10.3390/antibiotics12010072

**Published:** 2022-12-31

**Authors:** Isabella Letícia Esteves Barros, Flávia Franco Veiga, Lidiane Vizioli de Castro-Hoshino, Monique Souza, Amanda Milene Malacrida, Beatriz Vesco Diniz, Rafaela Said dos Santos, Marcos Luciano Bruschi, Mauro Luciano Baesso, Melyssa Negri, Terezinha Inez Estivalet Svidzinski

**Affiliations:** 1Postgraduate Program in Health Sciences, State University of Maringá, Colombo Avenue, 5790, Maringá 87020-900, PR, Brazil; 2Medical Mycology Laboratory, Department of Clinical Analysis and Biomedicine, State University of Maringá, Colombo Avenue, 5790, Maringá 87020-900, PR, Brazil; 3Department of Physics, State University of Maringá, Colombo Avenue, 5790, Maringá 87020-900, PR, Brazil; 4Laboratory of Research and Development of Drug Delivery Systems, Department of Pharmacy, State Unversity of Maringá, Colombo Avenue, 5790, Maringá 87020-900, PR, Brazil

**Keywords:** yeasts, biofilm, onychomycosis, biological products, propolis

## Abstract

Species of the *Candida* genus represent the third most common cause of onychomycosis, the most frequent and difficult to treat nail infection. Onychomycosis has been attributed to fungi organized in biofilm and some natural products have proved promising for its treatment. This study aimed to evaluate the antibiofilm activity of propolis extract (PE) and its by-product (WPE) on 7-day preformed biofilms produced by *Candida albicans* in polystyrene microplates, as well as in an ex vivo model on human nail fragments. The cytotoxicity and permeation capacity were also assessed. Firstly, multiple parameters were evaluated over 7 days to elucidate the dynamics of biofilm formation by *C. albicans*. The cell viability and total biomass did not vary much from the beginning; however, days 3 and 4 were crucial in terms of metabolic activity, which was significantly increased, and the levels of extracellular matrix components, wherein proteins and nucleic acids experienced an increase, but polysaccharide levels dropped. Architecturally, one-day biofilm showed a monolayer of organized cells (blastoconidia, hyphae, and pseudohyphae), while in the seven-day biofilm there was a three-dimensional well-structured and complex biofilm. This yeast was also able to form a biofilm on both surfaces of the nail, without an additional nutritional source. Both extracts showed excellent antibiofilm activity against the 7-day preformed biofilm and were not toxic to Vero cells at concentrations compatible with the antifungal and antibiofilm activities. Both extracts permeated the experimentally infected nail, with WPE being more efficient. The results of this study, taken together, reinforce the potential of these natural products, containing propolis, as a safe option for the topical treatment of onychomycosis.

## 1. Introduction

Among all the dermatomycoses, onychomycosis is one of the most feared and frequent fungal infections. In addition to causing social discomfort, onychomycosis can present as a variety of clinical lesions and, in many cases, this can involve physical pain and itching [1,2]. Several fungi can cause this infection and the yeasts of the genus *Candida* have come to represent the third most common onychomycosis agent [3,4]. Furthermore, *Candida albicans* is responsible for approximately 70% of all *Candida*-related onychomycosis cases [2,5]. It is already evident that the biofilm is an important fungal virulence factor involved in the etiopathogenesis of onychomycosis and that the organization of the fungi in this form justifies the characteristics of this infection [5,6,7]. Recently, we demonstrated that *C. albicans* produces a great amount of biofilm and that this was associated with the symptomatology easily observed in onychomycosis caused by this yeast, such as paronychia and pain [8].

The chronic nature of onychomycosis contributes to the high recurrence rates (ranging from 10% to 53% in the literature) [1,2,7] and the difficulty in treating infections due to resistance. Traditionally, oral antifungals have been the first choice of treatment because of their high success rates; however, they can be associated with drug interactions and can occasionally cause serious adverse effects [1,2]. Therefore, new therapeutic options have been investigated, such as the use of laser, photodynamic therapy, and topical use of natural products [1,7]. The antifungal properties of propolis on *Candida* isolates have been largely proved and other studies have already demonstrated the effectiveness of propolis extracts (PE) against yeasts of the genus *Candida* in mucosal infections [9,10,11,12,13,14,15] and also against onychomycosis isolates [16,17]. Recently, our group showed that a PE and a propolis by-product extract (WPE) have excellent antifungal activity in planktonic cells and great effectiveness in inhibiting the biofilm formation of three causative agents of onychomycosis, including *C. albicans* [18].

However, the dynamics of biofilm formation by *C. albicans* and its interaction with the human nail are still unclear. Furthermore, the antibiofilm and permeation ability of these two extracts (PE and WPE) have not been fully elucidated. Therefore, the objectives of this study were to produce an in vitro biofilm by *C. albicans* and characterize the properties related to biofilm formation over seven days, as well as to determine the ability of this yeast to form biofilm in an ex vivo model of human nail. The antibiofilm activity, cytotoxicity, and permeation capacity of PE and WPE were then evaluated in these models.

## 2. Results

### 2.1. In Vitro Evaluation of C. albicans Biofilm Formation over Seven Days

*C. albicans* biofilm was produced in flat-bottomed polystyrene microplates and various parameters were evaluated over seven consecutive days to determine the biofilm formation dynamics (Figure 1). The number of viable cells was practically constant over the 7 days (Figure 1A). The protein profile highlighted an increase in extracellular matrix (ECM) proteins on day 2, which remained stable until day 5, but was followed by decrease in levels on day six (Figure 1B). There was, however, a significant increase in the amount of biomass between days 1 and 2, followed by a significant decrease until the fourth day 4, after which it maintained a constant biomass (Figure 1C). A very similar profile was observed in the polysaccharides of the ECM (Figure 1D). In addition, the metabolic activity was found to be significantly more intense on the third and fourth days of biofilm formation (Figure 1E). A similar profile was observed in the nucleic acids, where there appeared to be an increasing trend in the nucleic acids present in the ECM, peaking at day 4, but there were no significant differences in the DNA and RNA levels between the time-points (Figure 1F). The SEM analysis showed that in the one-day biofilm (Figure 1G), the yeast had already started to form the structural organization of the biofilm, but still in a monolayer of cells (blastoconidia, hyphae, and pseudohyphae). In the seven-day biofilm (Figure 1H), a highly structured and complex biofilm, with a three-dimensional structure, was observed.

### 2.2. Assessment of the Ability of C. albicans to Form Biofilm in an Ex Vivo Model of Onychomycosis

*C. albicans* was then evaluated for its ability to form biofilm in an ex vivo model with human nail as the only nutritional source. The yeast was able to form biofilms on both sides of the nail, but growth on the ventral side was more intense and organized (Figure 2D,E). Furthermore, it was noted that there was intense growth of yeast in the fissures present on the nail surfaces (Figure 2C,F).

Analysis by Fourier transform infrared-attenuated total reflectance (FTIR-ATR) spectroscopy (Figure 3) showed that this yeast was able to infect the nail, regardless of the route of inoculation (dorsal or ventral), since there was an overlap of absorption peaks. Some components appeared in slightly higher amounts when the infection occurred on the ventral side, especially lipids (~2850–3000 cm^−1^). Furthermore, in both ventral and dorsal infections, there was an absorption peak at ~1050 cm^−1^, which was more intense on the side of the nail where the yeast was inoculated.

### 2.3. Antibiofilm Activity of PE and WPE against C. albicans

The minimum inhibitory concentration (MIC) of PE and WPE was determined previously [18] in terms of the total phenol content (TPC) of the extracts, and these concentrations were used as a basis for testing in the present model. Regarding the antibiofilm action of the extracts against the biofilms preformed in flat-bottomed polystyrene microplates for 7 days, there was a reduction of approximately 3 or 4 logs of CFU in *C. albicans* after exposure to WPE and PE, respectively (Figure 4A). The SEM images, likewise, demonstrated the effectiveness of both extracts in mature biofilms (Figure 4B), as a reduction in the amount of CFU could be observed visually, accompanied by a disorganization of the structures in the treated biofilms.

### 2.4. Cytotoxicity of PE and WPE

From the incorporation of the Neutral Red dye by viable cells, it was possible to calculate the cytotoxicity index (CC_50_) of PE and WPE in a mammalian cell line, Vero cells. The threshold of cytotoxicity found for PE and WPE was 3513.51 µg/mL of TPC and 313.08 µg/mL of TPC, respectively. These values indicated the concentrations of each extract, from which the cytotoxic effect would be relevant (50% induction of cell lysis or death). Regarding the selectivity index (SI), both PE (SI = 2.05) and WPE (SI = 2.28) were selective for yeast cells, with better performance for WPE.

### 2.5. Permeation Capacity of Both Extracts in an Ex Vivo Model of Onychomycosis Caused by C. albicans

The spectra of PE and WPE, before being applied to the nail, were obtained in the region of 400 to 750 nm by photoacoustic spectroscopy (PAS) and both showed broadband absorption below 700 nm (Figure 5A). The ex vivo model of nails infected with *C. albicans* on the dorsal or ventral surface and then treated on the dorsal surface resulted in the following optical absorption spectra: PE and WPE treatment (Figure 5B,C), respectively of nails after dorsal infection; and PE and WPE treatment (Figure 5E,F) of nails after ventral infection. Confirmation of the presence of extracts at the application site was made by reading the dorsal surface, which gave similar absorption spectra to those of extracts alone (filled markers, Figure 5B,C,E,F).

An increase in absorption could be noted in the region below 700 nm when comparing the spectra obtained from the ventral surface of the non-infected (control) and infected nails after application of the extracts with those obtained from the ventral surface of the untreated infected nail (empty square markers, Figure 2B,C,E,F). This increase confirmed that there was permeation, since the extracts were identified on the ventral surface reaching the infection site. In addition, the areas under the curve of the ventral spectra were integrated to quantify permeation (Figure 5D), and it was evident that permeation was better in ventrally infected nails and that WPE permeated the nails better than PE.

## 3. Discussion

As far as we know, no study to date has evaluated the dynamics of biofilm formation of *C. albicans* in vitro by recording its characteristics for seven consecutive days. Although the CFU remained constant over the evaluated period (Figure 1A), an increase in biomass and polysaccharides was observed in the initial days (Figure 1C,D), which can be attributed to the production of ECM and also to the development of the biofilm in a rich medium such as RPMI. The ECM composition is dynamic and can be affected by changes in growth conditions such as the composition of the growth medium, temperature, pH, etc. [19]. Following the increase in biomass, we noticed a peak in the metabolic activity on days 3 and 4 that coincided with peaks in the nucleic acid and protein level profiles of this biofilm (Figure 1B,E,F). One study that evaluated biofilm formed by *C. albicans* in a shorter time of just 48 h reported that proteins represent approximately 55% of the dry weight of the ECM, while e-DNA constitutes about 5% [20]. These authors also identified a total of 565 different proteins in the matrix, with distinct activities, such as enzymes involved in carbohydrate and amino acid metabolism, and also some enzymes potentially involved in matrix degradation, which are important for biofilm dispersion. Furthermore, the release of e-DNA in the matrix indicates mechanisms of cell lysis, quorum sensing, and excretion of DNA-containing vesicles, as it was recently demonstrated that the DNA present in the ECM of the *C. albicans* biofilm is largely composed of random non-coding sequences [19,20].

It was clear that in one day the yeast had already started the structural organization of the biofilm but was still only a monolayer of cells (Figure 1G). In seven days, it was possible to visualize a highly complex and organized biofilm, as there were more cell layers, and the fungal structures were rich in different forms (Figure 1H). The one-day biofilm illustrates the early stages of its formation, in which isolated cells adhere to the substrate to form a basal cell layer and proliferate, which involves filament formation. This is the initial step in which the yeast changes its morphology to filament and thereby acquires the ability to invade host mucosa or surfaces on inert medical devices [21,22].

*Candida* species have developed several virulence mechanisms that may be involved in the pathogenesis of onychomycosis. These yeasts produce extracellular hydrolases, including proteinases, phospholipases, and lipases that enhance the penetration of the pathogen into the host tissue [23,24]. It has been reported that when the culture medium is supplemented with keratin, the production of extracellular proteinase by these yeasts increases markedly [23]. In this way, keratin, which represents most of the composition of the nail, would act as an excellent growth environment for virulent strains of *Candida*. In the present study, it was evident that *C. albicans* is able to use the nail as the only nutritional source and form biofilm on both nail surfaces. In addition, growth from the ventral side was more intense and organized (Figure 2D,E). This ability may be associated with the opportunistic character of this yeast that “takes advantage” of the morphological characteristics of the nail to spread, since the ventral surface of the nail is more porous than the dorsal surface [25]. While the dorsal surface cells overlap to form a smooth surface, the ventral surface is irregular allowing for interdigitation with the nail bed. The ventral plate cells emerge from the nail bed making them an easy target for infections [25]. *C. albicans* exists as a commensal in the skin folds and subungual region of most individuals, which reinforces the ease of the nail infection from the ventral side.

Overlapping absorption peaks in FTIR-ATR spectroscopy indicated that *C. albicans* was able to establish infection in the nail, regardless of whether the inoculation occurred on the dorsal or ventral surface (Figure 3). However, it was possible to confirm that infection on the ventral surface was more efficient, as some components such as lipids (~2850–3000 cm^−1^) were present in slightly higher amounts, suggesting a tropism of the fungus to the ventral surface. In addition, in both ventral and dorsal infections, there was an absorption peak at ~1050 cm^−1^, initially identified as glycogen, but which may represent other organic compounds. Two studies using this type of spectroscopy also reported peaks in this spectral region (950–1085 cm^−1^) which were associated with polysaccharides, phospholipids, and nucleic acids produced by *C. albicans* both in the form of blastoconidia and hyphae [26,27]. Interestingly, this signal (~1050 cm^−1^) has been found in biofilms produced by *Fusarium oxysporum* during experimental ex vivo infection of human nails [28,29]. Thus, apparently these two opportunistic fungi appear to use similar mechanisms to cause nail infection.

Propolis has previously been shown to be effective against *C. albicans* biofilms, however in those studies the biofilms were only preformed for 24 h [11,30,31,32,33]. Here, we have shown that the profiles of certain biofilm properties are dynamic over 7 days and that both PE and WPE had antibiofilm activity against the mature, 7-day preformed biofilm. Both PE and WPE were thus demonstrated to be promising antibiofilm drugs, since the concentration of twice the MIC was enough to disorganize mature biofilm of *C. albicans*, reducing the number of CFU by 3 to 4 logs (Figure 4). This performance was better than what has been observed for most antifungals available on the market, which generally require much higher concentrations of drug to be effective against a fungus organized in the form of a biofilm [34].

Regarding cytotoxicity, both extracts evaluated were not toxic to Vero cells, at concentrations that were previously reported to have antifungal activity in *C. albicans* [18]. PE, at 1712.50 µg/mL of TPC, was able to inhibit the growth of planktonic cells of this yeast and also to reduce biofilm formation by 4 logs [18]. This concentration is at least twice lower than that corresponding to the CC_50_. Likewise, WPE, at a concentration of 137.50 μg/mL of TPC, was capable of inhibiting growth and reducing biofilm formation [18], which is almost 3 times lower than the CC_50_.These results together with the SI are important, as they show that the concentrations that could be used as a reference for treatment would act only on the fungus and not on mammalian cells. Few studies so far have evaluated the cytotoxicity of propolis-containing compounds on non-tumor cell lines. Some authors that used the same cell line as the present study reported different results regarding the toxicity; with a dose-dependent cytotoxic effect for another green propolis extract [35], and toxicity at specific concentrations for a red propolis extract [36]. As for WPE, this is the first study on Vero cells. The only study that also evaluated the cytotoxicity of an extract of this by-product was on two colon cancer cell lines, HT29-MTX and Caco-2, for which concentration-dependent viability also showing a concentration-dependent viability, ranging from 0.25 to 2.50 mg/mL, was observed [37].

The ability to permeate the nail is an important factor for any candidate for the topical treatment of onychomycosis. This is particularly crucial in the case of *C. albicans*, since we demonstrated a more intense and organized biofilm on the ventral surface of the nail and the drugs are usually applied on its dorsal side. Our results showed that both extracts were able to permeate the nail, mainly in those ventrally infected, with WPE being slightly more efficient than PE. But according to Figure 5, both were detected on the opposite nail face to that which received their application. These findings ensured the extracts reached the infection site and therefore were able to exercise their antifungal and antibiofilm activities. It is important to note that for this test, PE and WPE were applied in pure form, representing 8 to 16 times higher TPC concentration than the respective MIC. This is the way that they would be used in a possible treatment, further reinforcing the efficiency of both extracts. The better performance of WPE in diffusing through the nail was probably because this extract has less resin, since its dry residue content was shown to be lower than that found in PE [18,37]. In addition, other virulence factors of this yeast, such as the production of enzymes that facilitate tissue invasion, and the production and invasion of biofilm hyphae [23,24], cause changes in the nail that can also aid in the diffusion of extracts through it.

In this study, we showed the excellent activity of two extracts derived from propolis in mature biofilm artificially produced by *C. albicans* as well as the fact that both extracts have good diffusion in ex vivo models of onychomycosis. An evaluation of the antifungal activity of the extracts in biofilms formed on the nail to confirm that the activity is maintained in a more physiologically relevant model would be the next step for the continuation of the study with these extracts.

## 4. Materials and Methods

### 4.1. Fungal Strain

This study was conducted with a strain of *C. albicans* from the American Type Culture Collection (ATCC 90028). The yeast was cultured in CHROMagar™ Candida (Difco™, Detroit, MI, USA) to check the culture purity. Before all the experiments, *C. albicans* was subcultured in Sabouraud Dextrose Agar (SDA; Difco™, Detroit, MI, USA) overnight at 35 °C and the cellular density was adjusted following cell counts on a hemocytometer.

### 4.2. Biofilm Formation in Culture Microplates

This assay was based on the methodology described by Barros et al. [18]. Briefly, *C. albicans* was grown on SDA for 24 h at 35 °C, followed by inoculation in Sabouraud Dextrose Broth (SDB; Difco™, Detroit, MI, USA), which was then incubated for 18 h at 35 °C with shaking at 120 rpm. After incubation, the cells were harvested via centrifugation at 3000× *g* for 10 min at 4 °C, and then washed twice with 15 mL of sterile 0.1 M phosphate-buffered saline, pH 7.0 (PBS). The inoculum was adjusted to a final concentration of 1 × 10^7^ cells/mL in RPMI Medium 1640 (Gibco^®^, Grand Island, NY, USA) with l-glutamine (without sodium bicarbonate), supplemented with 2% glucose and 0.165 M 3-(*N*-morpholino) propanesulfonic acid (pH 7.2) as a buffer (Sigma–Aldrich^®^, St. Louis, MO, USA). Following this, 200 μL of the suspension was placed into the wells of a 96-well flat-bottomed microplate, which was then incubated at 35 °C in a shaker at 110 rpm, for 7 days. Every 24 h, 100 μL of RPMI was removed and an equal volume of fresh RPMI was added for renewal of the culture medium. All experiments were repeated on two occasions with individual samples evaluated in duplicate.

### 4.3. Determination of the Number of Viable Cells in Biofilm

The total number of viable cells for each day of biofilm formation was determined after recovery of the total yeast biomass [18]. First, each well was washed twice with sterile PBS. Then, 200 μL of PBS was added to each well and the biofilms were vigorously scraped with a pipette, which was transferred to a conical tube; this process was carried out five times, to a total of 1000 μL. The tubes were vortexed for 1 min, then subjected to 30% sonication for 50 seconds. Serial dilutions were performed in PBS, and 10 μL of each dilution was placed on an SDA plate that was incubated at 35 °C for 24 h. The number of cultivable cells was expressed as colony-forming units per milliliter (CFU/mL) and the results were presented in terms of log of CFU/mL.

### 4.4. Biofilm Biomass Quantification by Crystal Violet Staining

On each day, the biofilms in each well were washed three times with PBS and, after drying, 200 μL of methanol was added to each well for 15 min to affix the biofilms. Later, 200 μL of crystal violet (1% *v/v*) was added for 5 min. The wells were washed with sterile distilled water and 200 μL of acetic acid (33% *v/v*) was then added to dissolve the stain. The obtained solution was read in a microtiter plate reader (Orange Scientific, Braine-l’Alleud, Belgium) at 620 nm and the absorbance values were standardized per unit area of well (absorbance cm^−2^) [38].

### 4.5. Evaluation of Metabolic Activity by XTT Assay

The reduction assay of the tetrazolium salt, 2,3-(2-methoxy-4- nitro-5-sulphophenyl)-5-([phenylamino]carbonyl)-2H tetrazolium hydroxide (XTT; Sigma-Aldrich^®^, St. Louis, MO, USA), was used to determine in situ biofilm mitochondrial activity [38]. Following three washes with PBS, 120 μL of PBS, 40 μL of an XTT stock solution (500 μg/mL), and 40 μL of a phenazine methosulphate (PMS) stock solution (50 μg/mL) were added. The final concentrations of XTT and PMS in the wells were 100 and 10 μg/mL, respectively. The microplates were then incubated at 35 °C for 3 h, protected from the light. After this period, the absorbance of the obtained solution was read in a microtiter plate reader at 492 nm and the absorbance values were standardized per unit area of the well (absorbance cm^−2^).

### 4.6. Characterization of Biofilm

Total biomass was recovered followed by separation of the extracellular matrix (ECM) of the biofilms by mechanical methods of filtration through a 0.22 μm membrane (KASVI, São José dos Pinhais, PR, Brazil). Then, this filtrate containing the ECM was analyzed in relation to the quantification parameters of extracellular deoxyribonucleic acid (e-DNA), extracellular ribonucleic acid (e-RNA), total polysaccharides, and total protein based on the techniques described by Veiga et al. [28], with some modifications. The components were measured by optical density (OD) through spectrophotometry on a Nanodrop 2000™ (Nanodrop 2000 UV-Vis Spectrophotometer, Thermo Fisher Scientific, Waltham, MA, USA). ODs were read at 260 nm for nucleic acids, and the 260/280 and 260/230 ratios were used to estimate the concentration of total proteins and polysaccharides, respectively. For the negative control, the diluent (sterile PBS) was used.

### 4.7. Propolis and By-Product Extracts

Brazilian green propolis was obtained from an apiary of *Apis mellifera* L. bees, in the northwest of Parana state of Brazil (23°24′2″ S, 52°1′50″ W), and located inside a eucalyptus reserve, surrounded by native forest with a predominance of *Baccharis dracunculifolia* (Asteraceae). This research was registered in Brazil under SISGEN N° AC7A2F5. Propolis extract was prepared by turboextraction (30%, *w/w*; PE). Propolis by-product extract (WPE) was obtained during the preparation of PE. The physicochemical evaluation of both extracts has been previously described [18].

### 4.8. Antibiofilm Effect of Propolis and By-Product Extracts

To evaluate the action of PE and WPE, mature biofilms were produced as described in Section 4.2 [18]. Thereafter, the biofilms were treated with 200 μL of PE at 3425 µg/mL of total phenol content (TPC) and WPE at 275 µg/mL of TPC (twice the MIC concentration) as previously determined [18]) and the microplates were incubated at 35 °C for 24 h. Untreated controls (preformed biofilm incubated with 200 μL of RPMI 1640 medium) were also included. The number of cultivable cells was expressed as log of CFU/mL.

### 4.9. Effect of Propolis and By-Product Extracts on Mammalian Cells

To evaluate the cytotoxicity against mammalian cells, the Vero cell lineage, derived from African green monkey kidney (non-tumor cell), was used. For maintenance, DMEM medium (Sigma-Aldrich^®^, St. Louis, MO, USA) and RPMI 1640 plus 10% fetal bovine serum (FBS; Gibco^®^, Grand Island, NY, USA) were used. Incubation was carried out at 37 °C and 5% CO_2_.

Upon verifying at least 80% confluence in the culture flasks, the cells were resuspended using trypsin (Gibco^®^, Grand Island, NY, USA) and the cell concentration was adjusted to 2 × 10^5^ cells/mL in RPMI-10% FBS (200 µL) added to the wells of a microplate. After 24 h, PE and WPE were added in a sequence of 10 serially diluted concentrations ranging from 26.76 to 13,700 µg/mL of TPC for PE and 4.31 to 2204.5 µg/mL of TPC for WPE. The assay was performed in triplicate and incubated for a further 24 h at 37 °C and 5% CO_2_. Plate controls were performed and submitted to the same steps, where the positive control was cells that received no treatment and the negative control was culture medium only [39].

After this period, the compounds were removed, and the wells were washed with PBS then 20 μL of the Neutral Red solution (0.5 mg/mL; INLAB^®^, Diadema, SP, Brazil), diluted in DMEM without phenol red (LGC Biotecnologia^®^, Cotia, SP, Brazil), was added. After 3 h of incubation at 37 °C, the crystals were solubilized with an acid alcohol solution (1% *v/v* acetic acid in 50% absolute ethyl alcohol) and the reading was performed in a spectrophotometer (SpectraMax^®^ Plus 384, Molecular Devices, San José, CA, USA) at 540 nm. The values obtained were used to calculate the cell viability in each well and to determine the CC_50_, the concentration of the extract that induced 50% of lysis or cell death, according to the criteria of ISO 10993-5 [40]. The selectivity index (SI) of the extracts was obtained from the ratio between the CC_50_ values of mammalian cells and the previously determined MIC of yeast cells [18]. An SI value greater than or equal to 2.0 was considered important, as it indicated that the extract was at least twice as active in yeast cells as in mammalian cells [41].

### 4.10. Biofilm Formation and Permeance Analysis of Propolis and Propolis By-Product Extracts in Human Nails

#### 4.10.1. Human Nails

Nail fragments were collected from healthy adult volunteer donors and manually cut into equal-sized pieces before autoclaving at 121 °C for 20 min. The study protocol was approved by the Human Experimentation Ethics Committee under number 4.095.897/2020. For infection, 3 µL of a *C. albicans* suspension (containing 1 × 10^7^ cell/mL in 0.85% sterile saline) were carefully pipetted onto the ventral or dorsal surface of the nail surface. Infected nails were incubated for seven days at 35 °C, in a humid chamber to ensure biofilm formation. All experiments were performed in triplicate and repeated three times on different days.

#### 4.10.2. Fourier Transform Infrared-Attenuated Total Reflectance (FTIR-ATR) Spectroscopy

To evaluate the fungal–nail relationship in the ex vivo *C. albicans* infection of human nails, readings were taken from both surfaces of the nail, ventral and dorsal, according to Veiga et al. [28], after seven days of infection. Measurements were performed using a FTIR spectrometer coupled to a diamond crystal ATR accessory. Spectral range was 900–3000 cm^−1^ with 128 scans and a resolution of 4 cm^−1^. Data acquisition was performed via software on a computer connected to the spectrometer, with ATR and background correction.

#### 4.10.3. Photoacoustic Spectroscopy (PAS)

The evaluation of permeation of both extracts by PAS was performed as previously described [35]. First, the spectra of the extracts alone and the spectrum of the uninfected nail without the application of extracts (as a control) were obtained. Next, nail spectra were obtained 24 h after application of 2 µL of each extract on the dorsal surface of nails, in both non-infected (control) and dorsal or ventral surface-infected nails, as described above. The preparations were kept aseptically in a humid chamber at room temperature. After 24 h, readings were taken on the dorsal side and then on the ventral side to detect the optical absorption band of the extracts on both sides of the nail. To be indicative of permeation along the length of the nail, the extract bands should be detected on the ventral side, opposite to the side of application. All tests were performed in triplicate.

### 4.11. Scanning Electron Microscopy (SEM) Analyses

For SEM analysis, microplates containing the biofilms and the human nail fragments infected with *C. albicans* were washed twice with 0.1 M cacodylate buffer, dehydrated with alcohol (using 50% ethanol for 10 min, 70% for 10 min, 80% for 10 min, 95% for 10 min, and 100% for 20 min) and then air-dried. Prior to observation, the microplates with biofilms and the nails were mounted on gold-plated aluminum stubs and observed with a FEI Quanta^TM^ 250 scanning electron microscope (Leo, MA, USA) [38].

### 4.12. Statistical Analysis

All tests were performed in duplicate or triplicate and performed on two or three independent days. Data with a non-normal distribution were expressed as the mean ± standard deviation (SD). Significant differences among means were identified using the ANOVA test followed by Bonferroni multiple-comparison test. The data were analyzed using Prism 5 software (GraphPad, San Diego, CA, USA). Values of *p* < 0.05 were considered statistically significant.

## 5. Conclusions

The two extracts derived from propolis, particularly PE, demonstrated excellent activity in mature biofilm produced by *C. albicans*. Neither extract was considered toxic to Vero cells at concentrations compatible with antifungal activity and in the ex vivo model, both extracts were able to permeate the nail. Furthermore, as far as we know, this is the first study to evaluate the characteristics of a *C. albicans* biofilm for seven consecutive days, showing the variations in the evaluated parameters in a complete way. *C. albicans* was also able to use the nail as the only nutritional source and form a biofilm on both surfaces (dorsal and ventral) of the nail, although it was clear that the fungal growth on the ventral side was more intense and organized. These findings help to understand the pathogenesis of this fungus during onychomycosis infection and to better target its treatment. The results of this study, taken together, reinforce the potential of using these natural products as a safe option for the topical treatment for onychomycosis.

## Figures and Tables

**Figure 1 antibiotics-12-00072-f001:**
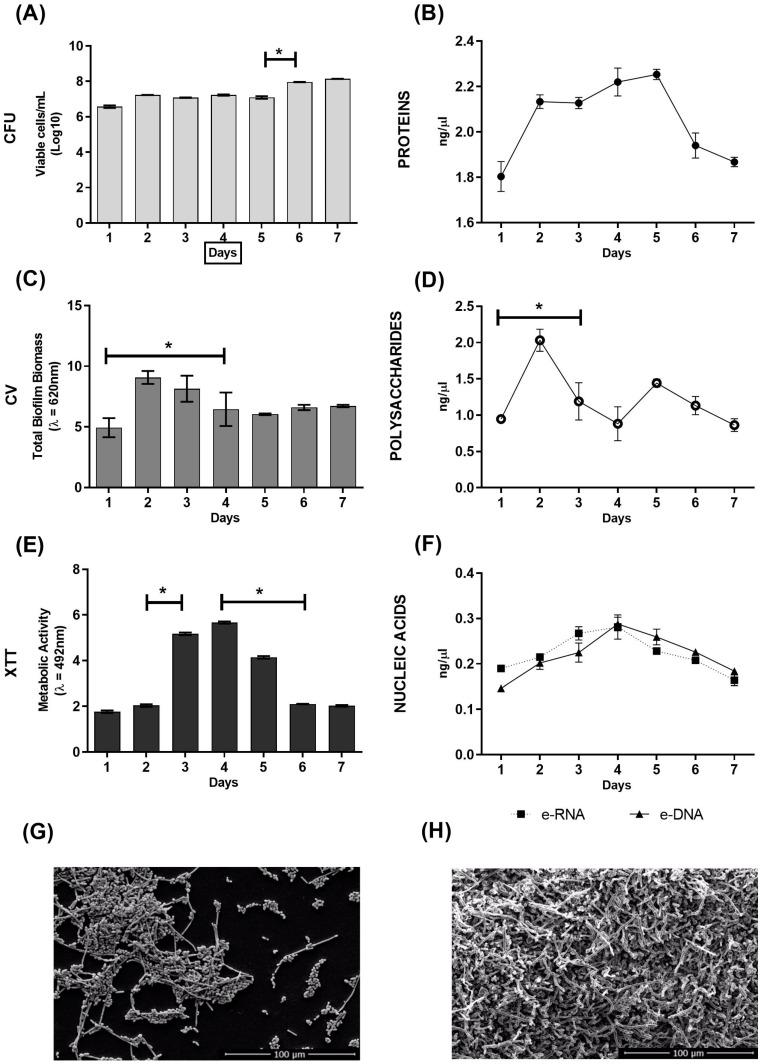
Dynamics of *C. albicans* biofilm formation on flat-bottomed polystyrene microplates over seven days. (**A**) Evaluation of cell viability by counting colony-forming units (CFU); (**B**) quantification of extracellular matrix (ECM) proteins; (**C**) quantification of total biomass by crystal violet (CV) staining; (**D**) quantification of ECM polysaccharides; (**E**) evaluation of metabolic activity by XTT reduction; (**F**) quantification of extracellular DNA (e-DNA) and e-RNA from the ECM; (**G**) morphological characterization by scanning electron microscopy (SEM) of the one-day growth of *C. albicans* biofilm; (**H**) morphological characterization by SEM of the seven-day growth of *C. albicans* biofilm. * Statistical differences between consecutive days. Graphs show the mean ± SD of 2 repeats and are representative of 2 independent experiments. Scale bar = 100 μm.

**Figure 2 antibiotics-12-00072-f002:**
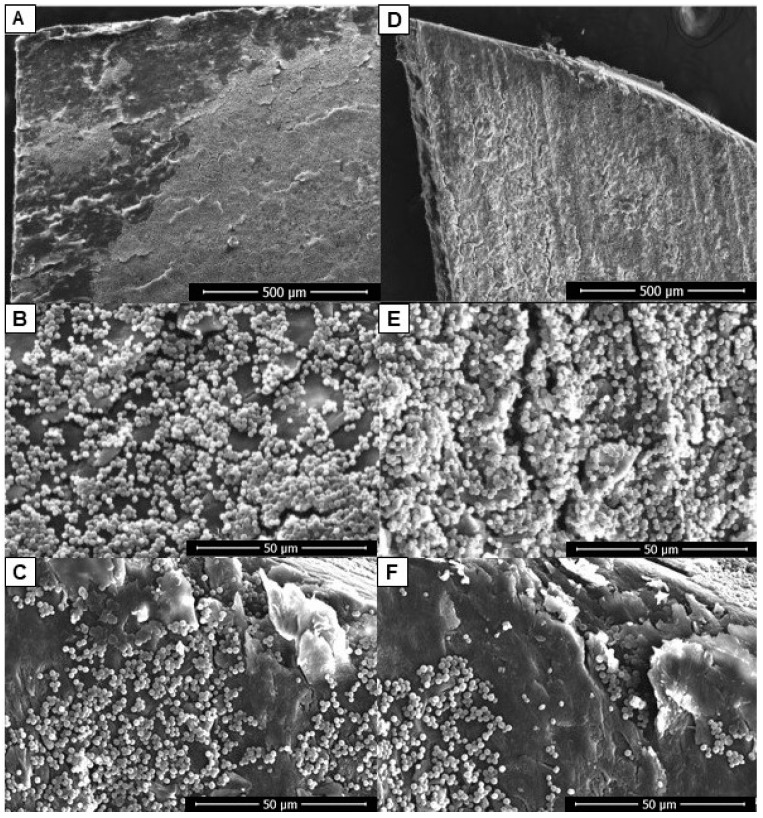
Biofilms of *C. albicans* on the nail’s surfaces in an ex vivo model of onychomycosis: dorsal (**A**–**C**) and ventral (**D**–**F**). SEM images showing the more intense and organized development of the fungus when applied on the ventral surface (**D**,**E**) in contrast to the more discreet growth on the dorsal surface (**A**,**B**). Intense yeast growth in the fissures present on the nail surfaces (**C**,**F**). Scale bar = 500 μm (**A**,**D**); scale bar = 50 μm (**B**,**C**,**E**,**F**).

**Figure 3 antibiotics-12-00072-f003:**
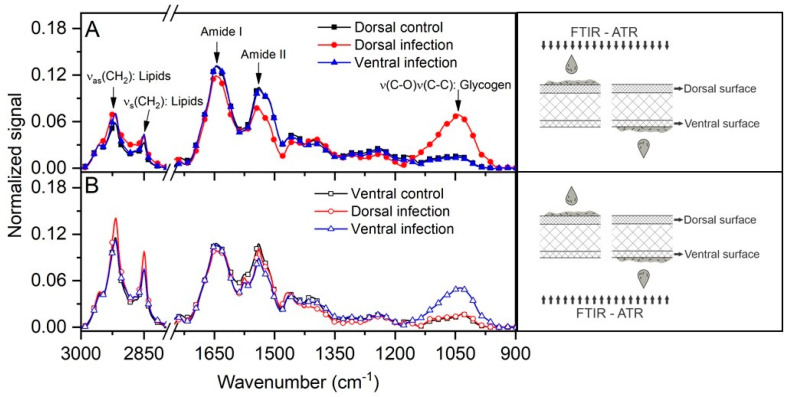
FTIR-ATR spectra analysis of the growth of *C. albicans* inoculated on both nail surfaces with readings obtained from the dorsal (**A**) and ventral (**B**) sides. Of note is the intensity of the peak at ~1050 cm^−1^, which was more intense on the side where the inoculation occurred. Readings were made according to the illustrative scheme presented on the right side of each graph.

**Figure 4 antibiotics-12-00072-f004:**
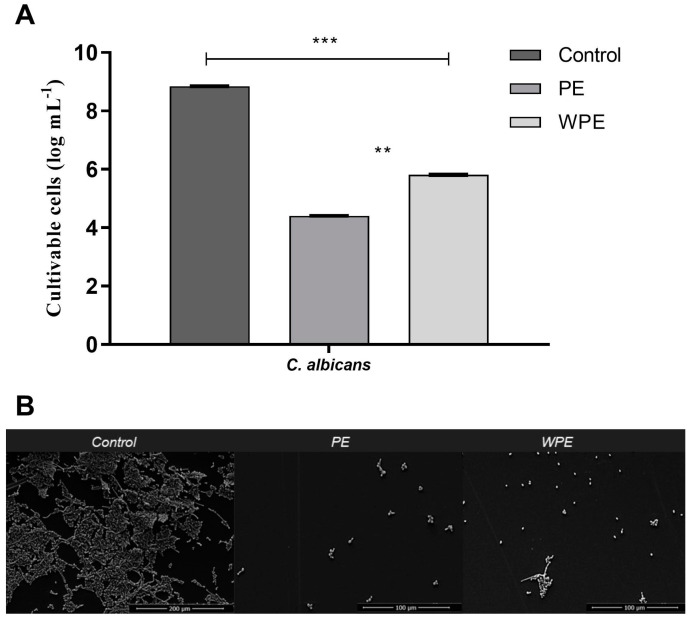
Effect of PE and WPE on seven-day preformed biofilm of *C. albicans* in flat-bottomed polystyrene microplates. Biofilms were treated for 24 h with PE and WPE at twice the minimum inhibitory concentration (MIC) which were 3425 µg/mL and 275 µg/mL of total phenol content (TPC), respectively. (**A**) Evaluation of cell viability by counting CFU. Statistical difference between: PE and WPE at *p* < 0.01 (**), and untreated control and treated biofilm at *p* < 0.001 (***). (**B**) SEM images of untreated (control) biofilms (magnification of 500; scale bar = 200 μM) and biofilms treated with PE and WPE (magnification of 1000; scale bar = 100 μM).

**Figure 5 antibiotics-12-00072-f005:**
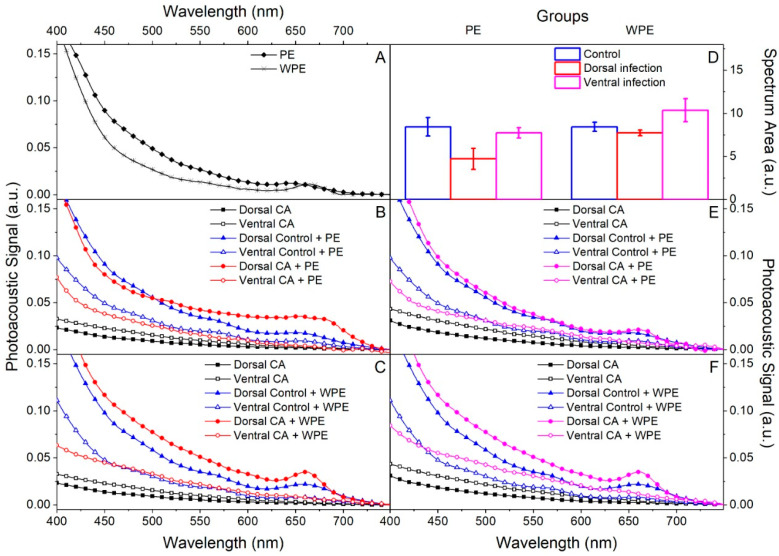
Evaluation of PE and WPE permeation of human nails by photoacoustic spectroscopy (PAS). Nails were infected for 7 days with *C. albicans* on the dorsal (**B**,**C**) and ventral (**E**,**F**) surfaces, then PE and WPE were applied once to the dorsal surface and the nails were incubated for 24 h. Readings obtained from the dorsal (filled markers) and ventral (empty markers) surfaces. (**A**) Readings of extracts alone. (**B**,**C**) Treatment with PE and WPE, respectively in non-infected nails (control—blue lines) and infected ones (red lines). Readings made in infected nails without extract (black lines). (**D**) Integration of the area under the curve of the ventral spectra to quantify permeation. (**E**,**F**) Treatment with PE and WPE, respectively, in non-infected nails (control—blue lines) and infected ones (pink lines). Readings made in infected nails without extract (black lines).

## Data Availability

Not applicable.

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
