# Peer review of "Performance of Two Extracts Derived from Propolis on Mature Biofilm Produced by Candida albicans"

_antibiotics, 2022, doi:10.3390/antibiotics12010072_

Round 1

Reviewer 1 Report

Τhe authors study the  performance of two extracts derived from propolis on mature  biofilm produced by Candida albicans.  In my opinion this project lacks originality and novelty, since there  is a large literature on the subject. In fact authors state that this is the case <<In fact, the efficacy of propolis on mature fungal biofilms of C. albicans has already been shown>> (line 231-232).  Ι would expect  a study with experiments trying to identify the biochemical or other mechanism of antifungal activity of the propolis and its  extracts that should be interesting for the readers. So, I propose that the study must be redesigned. 

Author Response

Reviewer

Τhe authors study the  performance of two extracts derived from propolis on mature  biofilm produced by Candida albicans.  In my opinion this project lacks originality and novelty, since there  is a large literature on the subject. In fact authors state that this is the case <<In fact, the efficacy of propolis on mature fungal biofilms of C. albicans has already been shown>> (line 231-232).  Ι would expect  a study with experiments trying to identify the biochemical or other mechanism of antifungal activity of the propolis and its  extracts that should be interesting for the readers. So, I propose that the study must be redesigned.

Author response & action taken:  We thank the reviewer because his comments provoked a deep reflection among the authors, since the news of this study was clear to us. There are at least 4 novelty in this study, which contribute to expanding knowledge about the etiopathogenesis and treatment of onychomycosis due to Candida albicans: 1) We have showed, by the first time, the characteristics of a C. albicans biofilm over seven consecutive days, with a proved efficacy of the propolis. 2) Both extracts, especially PE, demonstrated excellent activity on this kind of biofilm, which is more closed to real case of onychomycosis. 3) It was showed, by the first time, both extracts are not toxic to Vero cells. 4) both extracts were able to permeate the nail. 

We then realized that the text needed to be improved to make it clearer, thus we inform you that the manuscript has been completely rewritten.

Reviewer 2 Report

Barros et. al. reported the activity of two extract against candida albicans biofilm that could have potential therapeutic efficacy in nail infections. There are several issues of this study, particularly missing experiments and controls, that should be should done before considering the study for publication.

1)    What is the chemical composition of and major active ingredients PE and WPE and how they are different from each others?

2)    What is the aqueous solubility of the two extract and the antifungal mechanism of action?

3)    A positive control, using a FDA-approved antifungal agent, should be used in all antifungal and biocompatibility studies and in calculating the selectivity indices to be able to properly compare the efficacy of your antifungal agents.

4)    What is the concentration of PE and WPE used in antibiofilm study (Figure 4a)?

5)    Selectivity of indices of PE (SI= 2.05) and 137 WPE (SI= 2.28) are very low and basically show limited utility of these agents compared to know antifungal agents.

6)    In line 227, what is the MIC of PE and WPE?

7)    What is the dose-dependent toxicity of PE and WPE against Vero cells and other skin cells such as fibroblasts?

Author Response

Reviewer's Comments and Suggestions for Authors

Barros et. al. reported the activity of two extract against candida albicans biofilm that could have potential therapeutic efficacy in nail infections. There are several issues of this study, particularly missing experiments and controls, that should be should done before considering the study for publication.

1)    What is the chemical composition of and major active ingredients PE and WPE and how they are different from each others?

Author’s response: Regarding the chemical composition of the extracts, data have already been published in the previous article Barros, et al. 2022 (Promising effect of propolis and a by-product on planktonic cells and biofilm formation by the main agents of human fungal infections. An Acad Bras Cienc. 2022;94: e20210189. doi:10.1590/0001-3765202220210189). We reinforce that the by-product (WPE) is derived from the propolis extract (PE) itself, being the result of just one more extraction in ethanol, therefore the composition of both remains the same. What differentiates one from the other are the amounts of some actives present in each extract (data shown in the table below, published in the previous article). Basically, both extracts differ in terms of the amounts of dry residues and in the total phenol content (TPC) markers, already emphasized both in the previous publication and in the present study.

2)    What is the aqueous solubility of the two extract and the antifungal mechanism of action?

Author’s response: We inform you that the antifungal properties of propolis, including its mechanism of action, is a line of research in our laboratory from which numerous previous publications originated, highlighting here Correia et al. 2020 (Propolis extract has bioactivity on the wall and cell membrane of Candida albicans. J Ethnopharmacol. 2020;256: 112791. https://doi.org/10.1016/j.jep.2020.112791). As for the aqueous solubility of these extracts, we agree that it is a relevant parameter, but it was not part of the focus of the present study, since data on this approach, including the permeation capacity of both extracts in human tissue (nail) are in the phase for publication in another scientific journal.

3)    A positive control, using a FDA-approved antifungal agent, should be used in all antifungal and biocompatibility studies and in calculating the selectivity indices to be able to properly compare the efficacy of your antifungal agents.

Author’s response: We emphasize that in this study, no positive control was used with the interpretation that the reviewer made, as it was not an objective to compare results with a classic antimicrobial. In addition, standard strains were used, known to be susceptible to these commercial antimycotics.

4)    What is the concentration of PE and WPE used in antibiofilm study (Figure 4a)?

Author’s response: This concentration was informed in the methodology lines 383-384 

5)    Selectivity of indices of PE (SI= 2.05) and 137 WPE (SI= 2.28) are very low and basically show limited utility of these agents compared to know antifungal agents.

Author’s response: The SI indicates whether the sample is more selective for antifungal activity or more toxic for Vero cells. The more positive the SI value, the greater the selectivity to inhibit fungal growth; a negative value indicates that the sample is more toxic to Vero cells than selective for the inhibition of antimicrobial growth Martins et al., 2019 (https://doi.org/10.1155/2019/9423658). Furthermore, an SI value greater than or equal to 2.0 was considered significant to indicate whether the extract is twice as active in yeast cells as in mammalian cells. Thus, both tested PE or WPE products were twice as active for fungi than for mammalian cells.

Additionally, it is important to highlight that the selectivity index is directly related to the tested strain. A study conducted by Ferreira et al. 2020 (Ferreira, LS et al. SYNTHESIS, CHARACTERIZATION AND ANTIMICROBIAL ACTIVITY OF NEW Cu (II), Co (II) AND Sn (II) COMPLEXES WITH THE SODIUM HYDROTRIS(2-MERCAPTOTHIAZOLYL)BORATE LIGAND. Química Nova [online]. 2020, v. 43, n. 5, pp. 593-598. https://doi.org/10.21577/0100-4042.20170526.) showed SI for C. albicans and C. glabrata of 0.16 and 0.33, respectively for the antifungal ketoconazole, i.e., much lower values than we observed in our study, highlighting the efficiency of both PE and WPE.

6)    In line 227, what is the MIC of PE and WPE?

Author’s response: This information has been moved to line  133.

7)    What is the dose-dependent toxicity of PE and WPE against Vero cells and other skin cells such as fibroblasts?

Author’s response: The dose-dependent toxicity of PE and WPE against the colon cancer cell lines (Caco-2 and HT29-MTX) was added (line 278), against Vero cells were evaluated by the first time in the current study and the action of these extracts on fibroblasts, to our knowledge, has not yet been evaluated.

Reviewer 3 Report

The manuscript titled “Performance of two extracts derived from propolis on mature biofilm produced by Candida albicans” has the overall goal of assessing the cytotoxicity, antibiofilm activity, and permeation capacity of propolis extract (PE) as well as its by-product produced by Candida albicans on polystyrene, and its ability to form biofilm on the human nail surfaces using an ex vivo model. Results of the study show the possibility of using Pes as a safe option for the topical treatment of onychomycosis.

This manuscript requires some English language copyediting.

The introduction is also short and could be further substantiated.

There are so many past and existing studies on the use of PE on different species of Candida. I am unsure on what is the novelty of this study. Please see sample manuscripts below:

Antifungal activity of propolis on different species of Candida - Ota - 2001 - Mycoses - Wiley Online Library

Comparative study of in vitro methods used to analyse the activity of propolis extracts with different compositions against species of Candida - Sawaya - 2002 - Letters in Applied Microbiology - Wiley Online Library

The antifungal effect of six commercial extracts of Chilean propolis on Candida spp (scielo.cl)

Related to nail infection, there is one study below:

Yeasts as Important Agents of Onychomycosis: In Vitro Activity of Propolis Against Yeasts Isolated from Patients with Nail Infection | The Journal of Alternative and Complementary Medicine (liebertpub.com)

Please describe the importance of biofilm formation.

It is unclear as to what is the significance of seven days for the C. albicans biofilm evaluation.

Would there be variations in results as brought about by various ages of human subjects? What are the ages of the healthy adult volunteer donors and what is the rationale for choosing these subjects? How many subjects were involved?

Author Response

Comments and Suggestions for Authors

The manuscript titled “Performance of two extracts derived from propolis on mature biofilm produced by Candida albicans” has the overall goal of assessing the cytotoxicity, antibiofilm activity, and permeation capacity of propolis extract (PE) as well as its by-product produced by Candida albicans on polystyrene, and its ability to form biofilm on the human nail surfaces using an ex vivo model. Results of the study show the possibility of using Pes as a safe option for the topical treatment of onychomycosis.

This manuscript requires some English language copyediting.

Author’s response & action taken: We attended the recommendation. Amy Louise Goundry, an English native researcher, corrected the English language of this manuscript (certificate attached).

The introduction is also short and could be further substantiated.

Author’s response: A short introduction was indeed strategic, considering the long length of the other sections of this article, thus, aiming at an easy reading, we opted for a concise and sufficient introduction, since it contains 3 paragraphs addressing (general considerations, the problem with the gap and objectives).

There are so many past and existing studies on the use of PE on different species of Candida. I am unsure on what is the novelty of this study. Please see sample manuscripts below:

Author’s response: Yes, this subject has been extensively studied, including our research group, but we are sure about the novelties in this new publication: We emphasize that studies on the by-product (WPE) are little explored (2 previous articles) and this is a relevant novelty already that the proposal is to use a cheap material originated in a reuse of residues that would be discarded. Furthermore, the results presented in this article are unprecedented. Secondly, the results on the permeation of these two extracts on infected human nails are presented here for the first time. Finally, this study includes the differential of evaluating supposed antifungal drugs on mature (7days) Candida biofilm, since the other experiments are carried out involving newly formed biofilms (24 - 48h), but these times do not reflect the real situation of onychomycosis attributed by fungi organized in the form of very well established (mature) biofilms considered here for the first time evaluating at seven days. Therefore, we assume that such novelties are sustainable and relevant.

Antifungal activity of propolis on different species of Candida - Ota - 2001 - Mycoses - Wiley Online Library

Comparative study of in vitro methods used to analyse the activity of propolis extracts with different compositions against species of Candida - Sawaya - 2002 - Letters in Applied Microbiology - Wiley Online Library

The antifungal effect of six commercial extracts of Chilean propolis on Candida spp (scielo.cl)

Related to nail infection, there is one study below: Yeasts as Important Agents of Onychomycosis: In Vitro Activity of Propolis Against Yeasts Isolated from Patients with Nail Infection | The Journal of Alternative and Complementary Medicine (liebertpub.com)

Please describe the importance of biofilm formation.

 Author’s response: Currently, onychomycosis has been attributed to fungi organized in the form of a biofilm, as presented in the introduction lines 48-50 “It is already evident that the biofilm is an important fungal virulence factor involved in the etiopathogenesis of onychomycosis and that the organization of the fungi in this form justifies the characteristics of this infection [5–7]”. This question was the fundamental basis of the entire project from which the studies presented in this manuscript originated.

It is unclear as to what is the significance of seven days for the C. albicans biofilm evaluation. As mentioned in previous answers, the study of mature biofilms (seven days) becomes an important target in view of their association with the etiopathogenesis of onychomycosis. This theme was addressed in the introduction and discussion in this submitted manuscript.

Would there be variations in results as brought about by various ages of human subjects? What are the ages of the healthy adult volunteer donors and what is the rationale for choosing these subjects? How many subjects were involved?

Author’s response: We inform that the variables pointed out, with great propriety by the reviewer, were part of another study carried out by the same research group, which is in the publication phase and which concluded: fungi are able to grow on the human nail independently of the gender, age or side of the nail, besides Raman spectra showed slight differences in the nail composition from those volunteers. Therefore, these variables were not addressed in the present study, as they are considered irrelevant.

Round 2

Reviewer 1 Report

I appreciate that authors consider my comments constructive and that they led to the rewriting of the manuscript with more emphasis on the new findings, helping the reader to gain a clearer view of the subject. In the present form it  should be accepted for publication. 

Reviewer 2 Report

I recommend the paper for publication after evaluating the authors response and edited manuscript.